**Validation of Clyde River SuperDARN radar velocity measurements with the RISR-C incoherent scatter radar**

Alexander Koustov[1], Robert Gillies[2] and Peter Bankole[1]

[1]University of Saskatchewan, Saskatoon, Saskatchewan, Canada

[2]University of Calgary, Calgary, Alberta, Canada

Correspondence to: A.V. Koustov (sasha.koustov@usask.ca)

**Abstract**

The study considers simultaneous plasma velocity measurements in the eastward direction carried
out by the Clyde River SuperDARN high frequency (HF) radar and Resolute Bay incoherent scatter radar RISR-C. The HF velocities are found to be in reasonable agreement with RISR velocities up to magnitudes of 700-800 m s$^{-1}$ while for faster flows, the HF velocity magnitudes are noticeably smaller. The eastward plasma flow component inferred from SuperDARN convection maps (constructed for the area of joint measurements with consideration of velocity
data from all the radars of the network) shows the effect of smaller HF velocities more notably. We show that the differences in eastward velocities between the two instruments can be significant and prolonged for observations of strongly sheared plasma flows.

**1 Introduction**

The Super Dual Auroral Radar Network (SuperDARN) high frequency (HF) radars have been installed to continuously monitor the $\mathbf{E} \times \mathbf{B}$ plasma drift in the Earth's ionosphere (Greenwald et al., 1995). To achieve this goal, the radars detect coherent ionospheric echoes from the F region and measure their Doppler velocity. It is assumed that the decameter ionospheric irregularities, responsible for SuperDARN echoes, move with the velocity close to the $\mathbf{E} \times \mathbf{B}$ plasma drift. A
number of comparisons of SuperDARN velocity measurements with concurrently operating incoherent scatter radars (ISR) that measure the $\mathbf{E} \times \mathbf{B}$ plasma drift has been performed in the past (Ruohoniemi et al., 1987; Davies et al., 1999; Milan et al., 1999; Davies et al., 2000; Xu et al.,

2001; Gillies et al., 2009; Gillies et al., 2010; Bahcivan et al., 2013; Koustov et al., 2016; Gillies et al., 2018). These comparisons, overall, supported the above major assumption of the SuperDARN measurements. However, occasional significant differences between HF radar line-of-sight (LOS) velocities and $\mathbf{E} \times \mathbf{B}$ drift component along the beam have been noticed. Initially, these were thought to originate from differences in the echo collecting areas and in the signal integration time (Davies et al., 1999) but the body of the data published so far questions this notion. It is now accepted that the HF velocities of the F region echoes are generally smaller (Gillies et al., 2018). One factor found to lead to this result is an assumption made during SuperDARN velocity measurements, which sets the index of refraction for the ionosphere of unity. However, this explanation cannot account for large differences of more than 20-30%. Koustov et al. (2016) stressed the original finding by Xu et al. (2001) that the HF velocity magnitudes are substantially smaller (up to a factor of 2) than the $\mathbf{E} \times \mathbf{B}$ drift component for high-speed flows exceeding 1000 m s$^{-1}$. Furthermore, the HF velocity magnitudes are often larger than the $\mathbf{E} \times \mathbf{B}$ flow component along the radar beam (e.g. Ruohoniemi et al., 1987; Koustov et al., 2016; Gillies et al., 2018). Such observations have been interpreted in terms of lateral deviation of HF radar beams (Koustov et al., 2016; Gillies et al., 2018). Other SuperDARN-ISR velocity inconsistencies have been associated with the occurrence of E region echoes at traditionally expected F region ranges for SuperDARN (Bahcivan et al., 2013; Gillies et al., 2018).

Despite obvious progress in measurement interpretation, HF-based $\mathbf{E} \times \mathbf{B}$ measurements require further investigation if one wants to continue improving the quality of the convection mapping with HF radars. In addition, although all SuperDARN radars work on the same principle and often even have identical hardware, validation work for every unit is necessary to be confident in the reliability and consistency of measurements across the network.

In this study, we undertake validation study for the Clyde River (CLY) SuperDARN radar. In a broader context, this effort complements the previous validation work for the Rankin Inlet (RKN) and Inuvik (INV) SuperDARN radars by Koustov et al. (2009), Mori et al. (2012), Bahcivan et al. (2013), Koustov et al. (2016) and Gillies et al. (2018). Since the CLY radar currently provides a significant contribution to the global-scale convection mapping with SuperDARN such work is of particular importance. We take advantage of the availability of $\mathbf{E} \times \mathbf{B}$ plasma drift measurements made by the recently installed Resolute Bay (RB) Incoherent Scatter

Radar-Canada (RISR-C) (e.g. Gillies et al., 2016). In the present work, we compare CLY and ISR-based velocities in a different way than the previous studies.

Traditionally, gate-by-gate comparison of data from two radar systems that make measurements in roughly the same directions is performed (e.g. Gillies et al., 2018). Such an approach cannot be implemented for the CLY/RISR-C geometry because none of these radar's beams are close enough in terms of their direction (see map on Fig. 1 in Gillies et al. (2018)). For this reason, we consider RISR-C two-dimensional vectors in a certain area (which are inferred by merging data from multiple individual beams using the approach by Heinselman and Nicolls (2008)) and compare them with CLY data averaged over 3 beams and 4 gates. Thus, we assess the data in a statistical sense, in terms of the average and median velocities over a large spatial domain.

A validation using highly averaged data is appropriate since the SuperDARN global-scale maps of plasma flow obtained with the Potential Fit technique (Ruohoniemi and Baker, 1998; Shepherd and Ruohoniemi, 2000) are built using median-filtered LOS velocities (the so-called gridded velocities). These are inferred from up to 27 LOS velocity values in bins consisting of data in neighboring range gates ($\pm$ one) and radar beams ($\pm$ one) and for three consecutive radar scans. This implies that the input to the Potential Fit procedure is a highly smoothed HF velocity covering 3-6 min of raw data and a significant space domain. In this view, there is a sense in considering 2-D RISR-C data and compare them with HF velocity medians, or vectors from the convection maps, over large spatial areas of overlap.

Although our aim is to validate the CLY velocity measurements, there is additional value from the CLY-RISR comparison. The RISR method of velocity vector estimations also has some limitations (Heinselman and Nicolls, 2008) that need testing. A couple of the limitations we will consider are a lack of velocity measurements along magnetic field lines and the expectation of spatially quasi-uniformity of flows, which is not always satisfied. Thus, our work can be considered as a mutual validation of both radars' performance. Compatibility of the vector estimates by RISR and SuperDARN is expected, but the degree of this agreement is not yet known.

## 2 Geometry of RISR-C and Clyde River radar observations

Figure 1 shows the fields of view (FOV) of the CLY and RKN SuperDARN radars starting from range gate 5 and the location of the RB incoherent scatter radar RISR-C, which we will simply

refer to as the RISR radar hereafter. This radar makes measurements in multiple beams; it uses 11

beams in the so-called "world-day" mode and 51 beams in the so-called "imaging" mode.

Measured line-of-sight velocities in all the beams and at all ranges are used to infer 2-D vectors of

the $\mathbf{E}\times\mathbf{B}$ plasma flow according to the procedure outlined by Heinselman and Nicolls (2008).

The resultant vectors are reported with $0.25°$ bin size of magnetic/geographic latitude. The points

to which the measurements are assigned are shown in Fig. 1 for the height of 300 km. The actual

center line for the points of data merging depends on data availability in specific beams (Gillies et

al., 2018).

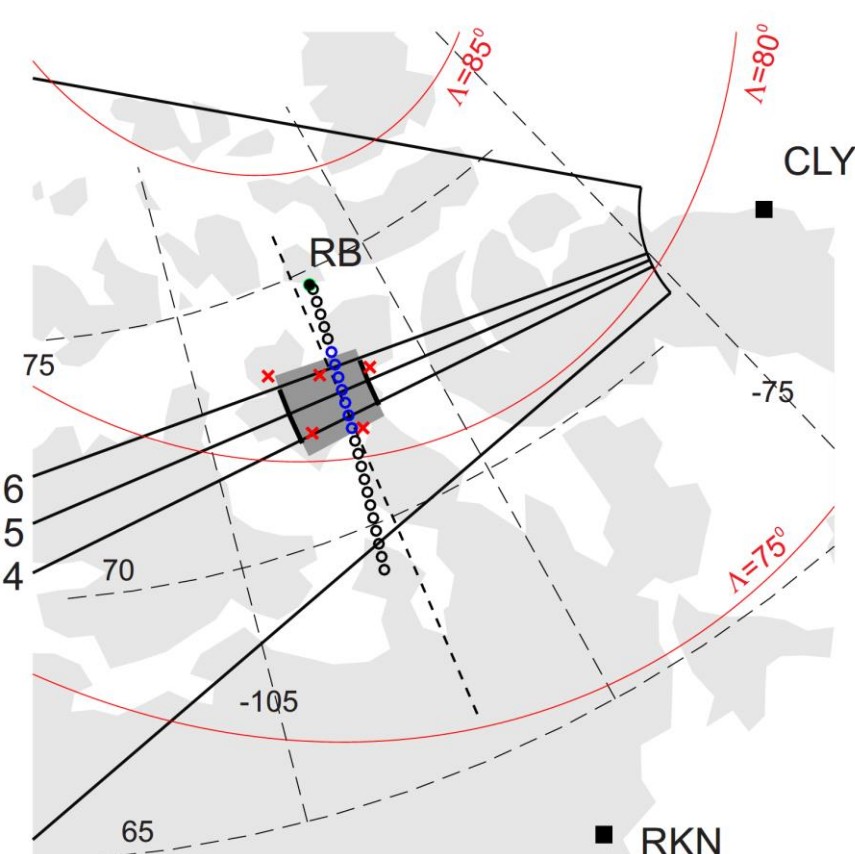

**Figure 1:** Field of view of the SuperDARN radar at RKN and CLY. The black straight lines are
the orientation of specific beams (4-6 for CLY) data from which were investigated. Shaded areas
represent areas of HF radar data averaging. RB is the location of the RISR-C ISR. The radar
reports $\mathbf{E}\times\mathbf{B}$ vector with a bin size of $0.25°$ of geographic latitude for points shown as black
circles stretching roughly along the magnetic meridian crossing the RB zenith. The blue-colored
circles are those locations whose data were used for comparison with the CLY measurements. The
solid red arcs are the magnetic latitudes of $75°, 80°$ and $85°$.

Figure 1 also shows the orientation of the CLY beams 4, 5, 6 (along their centers) and the area from which data were considered, the shaded rectangle region flanked by beams 4 and 6 between range gates 18-22. The monitored ionospheric region is centered at geographic latitude of $\sim 72.5°$. An important feature of this area is that within this range gates the CLY beams 4-6 are almost parallel to the lines of equal geographic latitude at the chosen radar range gates, as shown in Fig. 1. This means that one can directly compare CLY LOS velocities with the eastward component (in geographic coordinates) of a RISR $\mathbf{E} \times \mathbf{B}$ velocity vector. We note that the area of CLY observations was also covered by measurements from the RKN and INV radars (and occasionally by the Saskatoon and Kodiak SuperDARN radars), so that SuperDARN convection maps were usually well constrained.

**3 Methodology of the LOS velocity comparison**

We consider here an extensive data set comprising of about 1,000 hours of RISR measurements made over the entire year of 2016. On the days when the radar was operational it typically worked for the whole 24 hours, albeit switching, once-in-a-while, its mode of operation, except the world-day mode which usually covered an entire day. The range resolution of measurements in both modes is ~50 km. The data are available for winter and both equinoxes, with no measurements made in the summer. We consider 5 min RISR data because they have much smaller errors than the 1 min data that are also available.

Our approach to the CLY-RISR velocity comparison is as follows. We first select a 5 min period of RISR velocity measurements at geographic latitudes of $\sim 71.625° - 73.125°$ (see blue circles in Fig. 1) and compute the median velocity value for RISR. We then compute the median value of the CLY velocity over matching 5 minute interval in 3 beams and 4 gates, as mentioned above. The matched data pair is then entered into a common dataset.

Figure 2 shows the total number of 5 min intervals of joint RISR-CLY radar measurements, times when RISR and CLY both made measurements in the blue and shaded regions shown in Fig.1, as a function of UT. This histogram distribution does not include individual events when CLY data were obviously contaminated by ground scatter profoundly affecting the velocity comparison (Gillies et al., 2018). The ground scatter was identified with the conventional selection criteria (e.g. section 4.1 in Ponomarenko et al., 2007). The number of intervals was much larger

from noon to dusk (local solar noon is at about 19:00 UT).This is because of the preferential occurrence of CLY echoes at ranges of interest during the daytime (Ghezelbash et al., 2014).

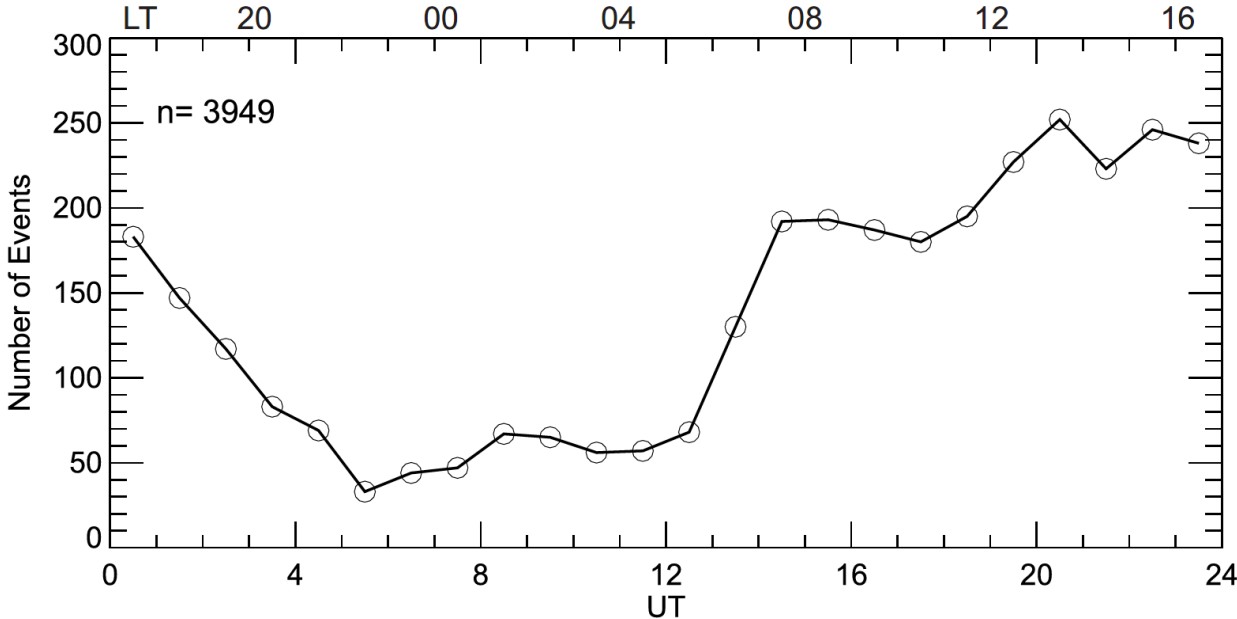

**Figure 2:** Number of CLY/RISR 5 min intervals of joint observations for all data considered. Total number of available intervals is shown in the top-left corner. For the area of observations, local time (scale at the top) roughly coincides with the magnetic local time.

## 4  Results for CLY LOS velocity - RISR comparison

Figure 3a shows the CLY LOS velocity versus the RISR Eastward $\mathbf{E} \times \mathbf{B}$ component for the entire dataset, produced as described above. The total number of points is close to 4000, which is a significant number. Overall, both positive and negative velocities are well represented. Although some spread is present, a significant amount of points are located close to the line of equality. To assess the plot, we binned the data according to the RISR measurements by using $100 \, \mathrm{m \, s^{-1}}$ bins of

the latter. Binned in this way CLY velocity medians are shown by black-white dots. The vertical black-white bars crossing each dot are the binned CLY velocity value $\pm$ one standard deviation. We also binned the data of Fig. 3 according to bins of the CLY velocity, pink asterisks (shown by thin symbols in order not to contaminate the plot).

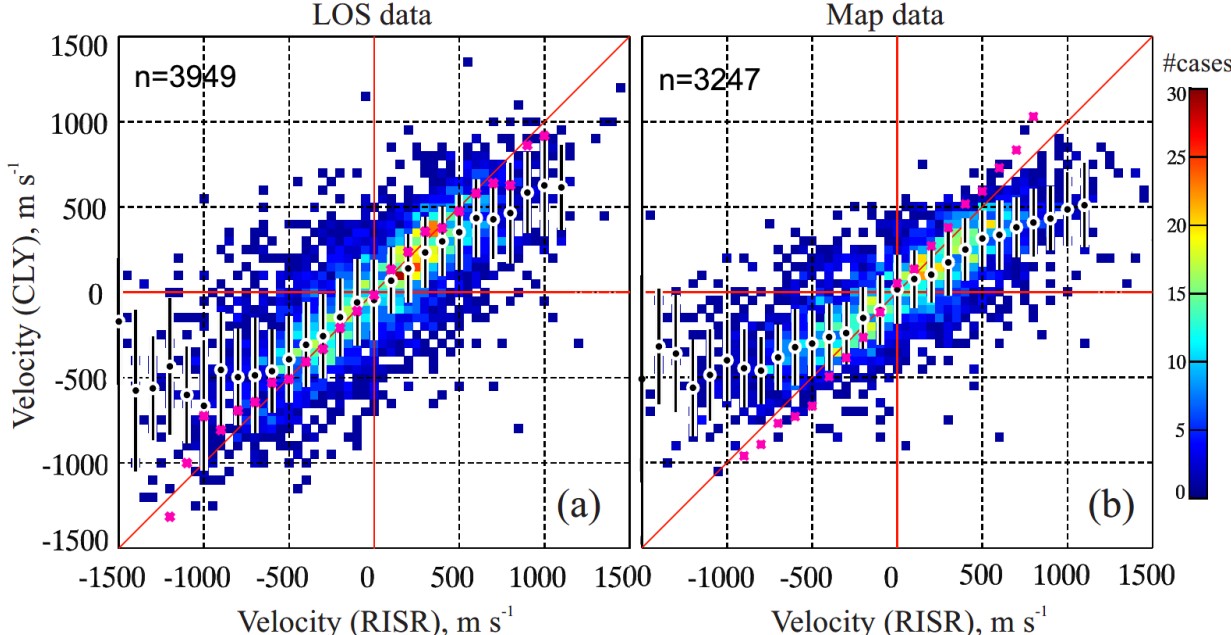

**Figure 3:** (a) Scatterplot of the CLY LOS velocity versus **E**×**B** eastward velocity component as inferred by RISR. Total number of points *n* is shown in the top left corner. The black-white dots are medians of the CLY velocity in $100\,\mathrm{m\,s^{-1}}$ bins of RISR velocity. The black vertical lines are the standard deviations of the CLY velocity in each bin. The pink dots are medians of the RISR velocity in $100\,\mathrm{m\,s^{-1}}$ bins of CLY measurements (b) The same as (a) but the eastward flow component inferred from SuperDARN flow maps was considered.

The black-white dots are reasonably close to the line of the perfect agreement.The pink asterisks are actually very close to the line of equality. Good alignment with the line of equality and good correspondence between the location of the black dots and pink asterisks indicate that the velocities are almost linearly related, especially in the range from -500 $\mathrm{m\,s^{-1}}$ to +500 $\mathrm{m\,s^{-1}}$. One clear departure of the back dots from the line of equality are the RISR velocities with magnitudes greater than ~750 $\mathrm{m\,s^{-1}}$.

An alternative way of assessing the data trends in Fig. 3a is to make a linear fit to the cloud of points. Parameters of the linear fit are presented in Table 1 for four ranges of the RISR velocity, $\pm500\,\mathrm{m\,s^{-1}}$, $\pm750\,\mathrm{m\,s^{-1}}$, $\pm1000\,\mathrm{m\,s^{-1}}$ and $\pm1500\,\mathrm{m\,s^{-1}}$. The slope is 0.73 for the smallest velocities of $\pm500\,\mathrm{m\,s^{-1}}$, which includes about 71% of all the data points. A linear fit to almost all the data has a decreased slope of 0.64.

Table 1. Parameters of the linear fit line $\text{Velocity}_{CLY}=a\cdot\text{Velocity}_{RISR}+b$, the number of points involved in the fitting and the squared correlation coefficient for various ranges of the RISR velocity. Left 3 columns are the LOS velocity comparison while right 3 columns are for the 2-D velocity comparison.

| | LOS comparison | | | | 2-D comparison | | | |
|---|---|---|---|---|---|---|---|---|
| $\text{ms}^{-1}$ | a | $b(\text{ms}^{-1})$ | Points | $R^2$ | a | $b(\text{ms}^{-1})$ | Points | $R^2$ |
| ±500 | 0.73 | -9.90 | 2815 | 0.46 | 0.59 | -16.32 | 2202 | 0.45 |
| ±750 | 0.71 | -10.29 | 3558 | 0.58 | 0.56 | -11.31 | 2851 | 0.57 |
| ±1000 | 0.68 | -8.27 | 3823 | 0.6 | 0.54 | -9.91 | 3106 | 0.62 |
| ±1500 | 0.64 | -5.31 | 3932 | 0.6 | 0.502 | -7.08 | 3227 | 0.61 |

**5 Methodology of "vector" comparison between SuperDARN and RISR**

The approach to the velocity vector comparison between the RISR and SuperDARN data is as follows. We restrict consideration to the same area of joint CLY-RISR observations as in the LOS comparison, shown in Fig. 1. Here the SuperDARN convection vectors are available at geomagnetic latitudes of $80.5°-81.5°$ and $\sim 7°$ of magnetic longitude. In this area, the convection maps/vectors are mostly based on RKN, INV and CLY radar measurements with only occasional contributions from other SuperDARN radars. We selected the three grid nodes at $81.5°$ magnetic latitude that were closest to the area of the CLY LOS velocity assessment and the two closest grid nodes at $80.5°$ magnetic latitude, marked by red crosses in Fig. 1. For each vector location, the geographic East component of the flow was computed and the median value (out of potentially 5 values, although for some periods it was as low as 1 measurement) was calculated to represent the eastward plasma flow component of a 5 min SuperDARN map. This is not a traditional temporal resolution for the SuperDARN mapping (which is usually 2 min); such data processing has been done to avoid the need of additional averaging of 2 min SuperDARN maps. Unfortunately, the start times of RISR measurement intervals were often irregularly spaced while SuperDARN maps were synchronized to exactly correspond to 5 min boundaries (i.e., 0-5 min, 10-15 min, etc.). For

the comparison, only HF and ISR data that were less than 2 min apart were considered. For this

reason, even when both radar systems were operational, the actual number of joint points per hour

was below the expected number of 12.

For RISR, the eastward $\mathbf{E} \times \mathbf{B}$ plasma velocity component was usually available at all points shown by open circles in Fig. 1. For the comparison with SuperDARN vectors, only measurements at geographic latitudes between $71.625° - 73.125°$ (given with a bin size of $0.25°$

, blue-colored circles in Fig. 1) were considered, and the median value of the eastward component was computed. The selection criteria produced a slightly shorter (but still statistically significant) data set than was obtained for the LOS velocity comparison. We stress that although the data for the comparison were along one specific direction, geographic east, two-dimensional vectors were used in determination of the velocity component for both systems with the SuperDARN vectors

calculated using measurements from all radars including CLY, RKN and INV, as well as the statistical model by Ruohoniemi and Greenwald (1996).

**6 Results for "vector" comparison between SuperDARN and RISR**

Figure 3b plots eastward component of the plasma flow measured by RISR and SuperDARN. The spread of the data looks similar to that of Fig. 3a (the LOS comparison). We assessed Fig. 3b using the same methods as performed on Fig. 3a (see section 4). Overall agreement of the data clearly holds.

Several results from Fig. 3b are consistent with the data of Fig. 3a. First, the SuperDARN

map-based velocities are somewhat smaller than those of RISR. This is recognizable through an obvious deviation of the distribution maxima from the line of equality, especially at RISR positive velocities of $> 500$ m s$^{-1}$. Secondly, the tendency for the SuperDARN velocity being smaller is greater for larger RISR magnitudes. This feature is seen for both positive and negative RISR velocities. Finally, consistent with previous reports (Koustov et al., 2016; Gillies et al., 2018),

there is a number of points for which the radars show oppositely directed flows. This was more frequent for small RISR velocities. Although Fig. 3 shows good consistency of the data provided by the two radar systems, the differences can be as large as a factor of 2 in individual measurements.

The agreement between the convection vectors given by RISR and SuperDARN is expected. We see that the consistency deteriorates once 2-D data are involved, but mostly at intermediate velocity magnitudes of 300-600 m s$^{-1}$. The inconsistencies are characterised by slower SuperDARN velocities. Interestingly, the differences for large velocity magnitudes in Fig. 3b are comparable to those in Fig. 3a.

To assess the data trends in Fig. 3b in alternative way, linear fits to the scatter of points in Figure 3b were made for four ranges of the RISR velocity of $\pm500\,\mathrm{m s}^{-1}$, $\pm750\,\mathrm{m s}^{-1}$, $\pm1000\,\mathrm{m s}^{-1}$ and $\pm1500\,\mathrm{m s}^{-1}$, similar to those for the LOS velocity comparison. The slope of the fitted line, the y-intercept, the number of points involved in each fitting and the squared correlation coefficient are presented in Table 1. The slopes are close to 0.6 for the set of smallest velocities ($\pm500$ m s$^{-1}$), which includes about 68% of all the available data. The slope decreases to 0.5 if almost all the data are considered. We think that the deterioration of the agreement at intermediate and large velocity magnitudes is due to the broader area over which the SuperDARN data are averaged for the 2-D comparison. In this case, there is more chance for SuperDARN to include ground-scatter contaminated measurements, giving effectively slower grid velocities to the fitting procedure.

## 7. On possible reasons for velocity disagreements

One reason frequently given for the systematic "underestimation" of the SuperDARN velocity measurements is the assumption that the index of refraction is unity (Gillies et al., 2009; Ponomarenko et al., 2009). We attempted to evaluate the importance of this effect in our data set. A plot similar to Fig. 3a was produced, but with the CLY velocity being corrected by considering the electron density (at the F region peak) measured by RISR. The plot looked very much similar to Fig. 3a. We assessed the plot by applying the linear fit line to the HF velocity medians in 100 m s$^{-1}$ bins of RISR velocity, considering the range of almost linear dependence, between -1000 and +1000 m s$^{-1}$ of RISR velocities. The slope of the best fit line improved to ~0.75 (from ~0.65). This improvement is consistent with the previous studies though it does not entirely account for the differences between the radar measurements.

We also investigated the diurnal variation of the velocity ratio $R = Vel_{HF}/Vel_{RISR}$ as done previously by Gillies et al. (2018) to explore possible influences of the refractive index on velocity using typical local time variations in the electron density as a proxy for refractive index. For the winter and equinoctial ionosphere over Resolute Bay, the largest densities are systematically observed near local solar noon and during the afternoon hours (18-22 UT) (e.g. Ghezelbash et al., 2014; Themens et al., 2017). It is therefore expected that the velocity ratio $R$ would be smallest during these times, as reported by Gillies et al. (2018) for the RKN radar. The nighttime results by Gillies et al. (2018) are more confusing. First, strangely, the ratios here were often above 1 at latitudes southward of RB and systematically below 1 (but not as far below unity as they were near noon) at latitudes poleward of RB. Gillies et al. (2018) indicated that the vertical plasma flow velocities in RISR measurements were, very likely, incorrectly estimated for nighttime observations. Since the observation area in our comparison is close to RB, we expect that this effect will also affect the RISR-CLY comparison.

Figure 4 plots the hourly median ratio $R$ as a function of time for our CLY-RISR data set. One can see that $R$ varies significantly. It is lower during daytime (noon is at about 19:00 UT) than during dawn/prenoon (12-18 UT), but its values are smallest during nighttime (midnight is at about 07 UT). Interestingly, the average ratio over all UTs is 0.83, which is closer to 1 than the slopes of the lines in Figs. 3 (Table 1). This is probably because the infrequent high-velocity data are averaged out by dominating data at low velocities in certain RISR bins of Fig. 4.

We think that the low nighttime $R$ values are caused by "overestimation" of true plasma drift in a plane perpendicular to the magnetic field by RISR in the midnight sector. We note that this is not quite consistent with Gillies et al. (2018) who interpreted their nighttime data in terms of effectively decreased RISR LOS velocities. Our data suggest effectively increased RISR velocity magnitudes. One of the factors affecting the derivation of the "averaged" flow pattern at nighttime, for both radar systems, is that the flows in this sector are often very irregular, even in the polar cap (Bristow et al., 2016). Under these conditions, the solution is subject to large uncertainty.

Gillies et al. (2018) believe that large nighttime ratios of RKN to RISR velocity could be due to errors in HF measurements because the RKN beams can experience significant lateral deviations so that actual measurements are performed at smaller flow angles with a larger LOS

velocity component. This explanation cannot be applied to our observations. This is because the CLY radar observes azimuthally, along the average plasma flow most of the time (except of short periods at near noon and near midnight when the flows are predominantly meridional) so that lateral deviations of the CLY beams would lead to, depending of the orientation of the plasma flow with respect to the CLY beam, either smaller or larger LOS velocities.

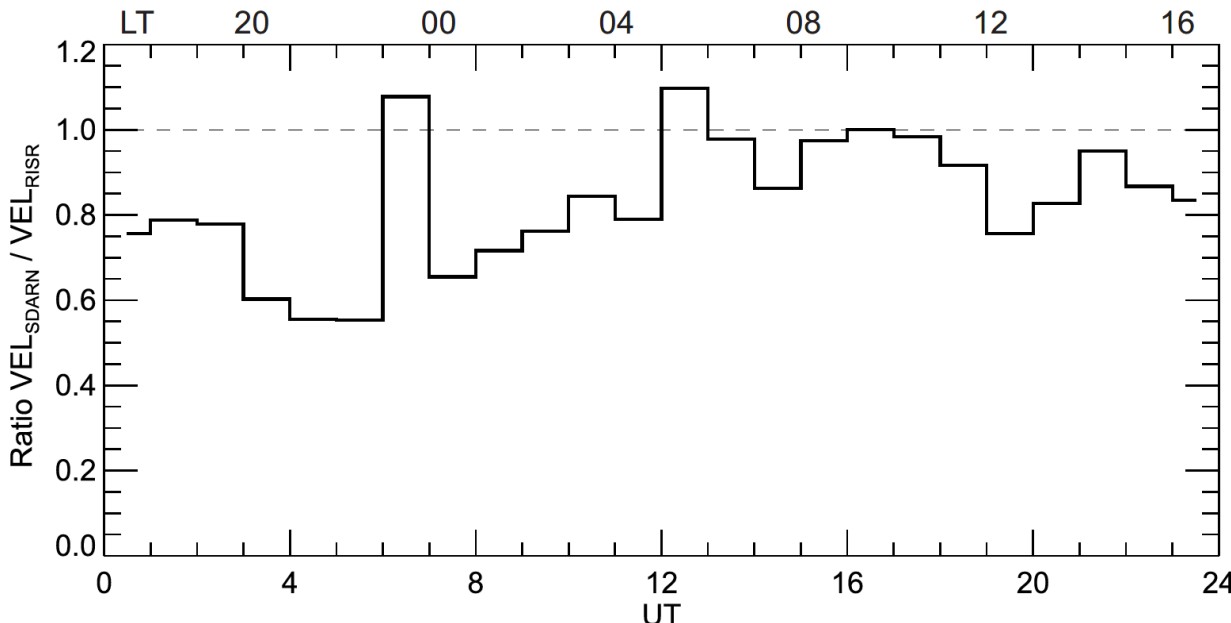

**Figure 4:** Line plot of the hourly median velocity ratio *R* versus UT for the CLY radar. The data set is the same as for Fig. 3a. For the area of observations, local time (scale at the top) roughly coincides with the magnetic local time.

We think that the HF-RISR velocity inconsistency can also originate, at least partially, from the nature of HF signal formation. The effect has been discussed in general terms by Uspensky et al. (1989) as applied to E region coherent scatter and by Koustov et al. (2016) for F region coherent backscatter. The flows in the nighttime ionosphere are very likely to be more patchy/grainy with occasional occurrence of regions with enhanced flow magnitude (low electron density) and decreased flow magnitude (high electron density). We argue that in the case of a patchy ionosphere, there is a good chance that the ratio *R* would be smaller than in the case of a uniform ionosphere and homogeneous flow. Flow enhancements and decreases affect both RISR and HF measurements but in a profoundly different manner. The RISR radar would average the

velocity in patches with enhanced and depleted electron density together, and it would report what

can be classified as the "background" flow velocity. In the presence of electron density patches with enhanced and decreased $\mathbf{E}\times\mathbf{B}$ plasma flows, HF radars would preferentially detect stronger signals from those areas where the electron density is enhanced, and the electric field (flow magnitude) is decreased, so that they would show somewhat smaller velocity than the background value measured by an incoherent scatter radar.

It is conceivable to have the opposite situation with HF velocities above the background flow if regions with enhanced density have stronger local electric field, as discussed in Uspensky et al. (1989). In this respect, Koustov et al. (2016) and Gillies et al. (2018) noticed that HF velocities could be larger than the $\mathbf{E}\times\mathbf{B}$ plasma drift component measured by ISRs. Such points are occasionally seen in previously published data (Ruohoniemi et al., 1987; Davies et al., 1999).

Our data in Figure 3 also show such points but, in general, the data agree fairly well. Although the work of Koustov et al. (2016) and Gillies et al. (2018) related the larger HF velocity effect to lateral deviations of the HF radar beams from the expected directions, it could partially be due to the aforementioned effect of ionospheric microstructuring.

Potentially, low $R$ values can be related to the occurrence of misidentified ionospheric

scatter because some ionospheric echoes with low velocities can actually be ground or mixed ionospheric and ground scatter. Gillies et al. (2018) showed that removal of points that could potentially be affected by ground scatter improves the RKN-RISR velocity agreement significantly. Our analysis showed that ground scatter is rare during winter/equinox nighttime for the CLY radar which is consistent with low nighttime F region densities (Ghezelbash et al., 2014;

Themens et al., 2017). We also have to remind the reader that presumably obvious events with CLY ground scatter contamination have been removed from our consideration in Fig. 3.

Investigating our database, we identified one special situation when the RISR-SuperDARN velocity disagreements were particularly strong. Figure 5 gives an example of CLY-RISR observations on 4 March 2016 where RISR and CLY velocities differ consistently by several

hundred $\mathrm{m\,s^{-1}}$ over a period of almost 2 hours.

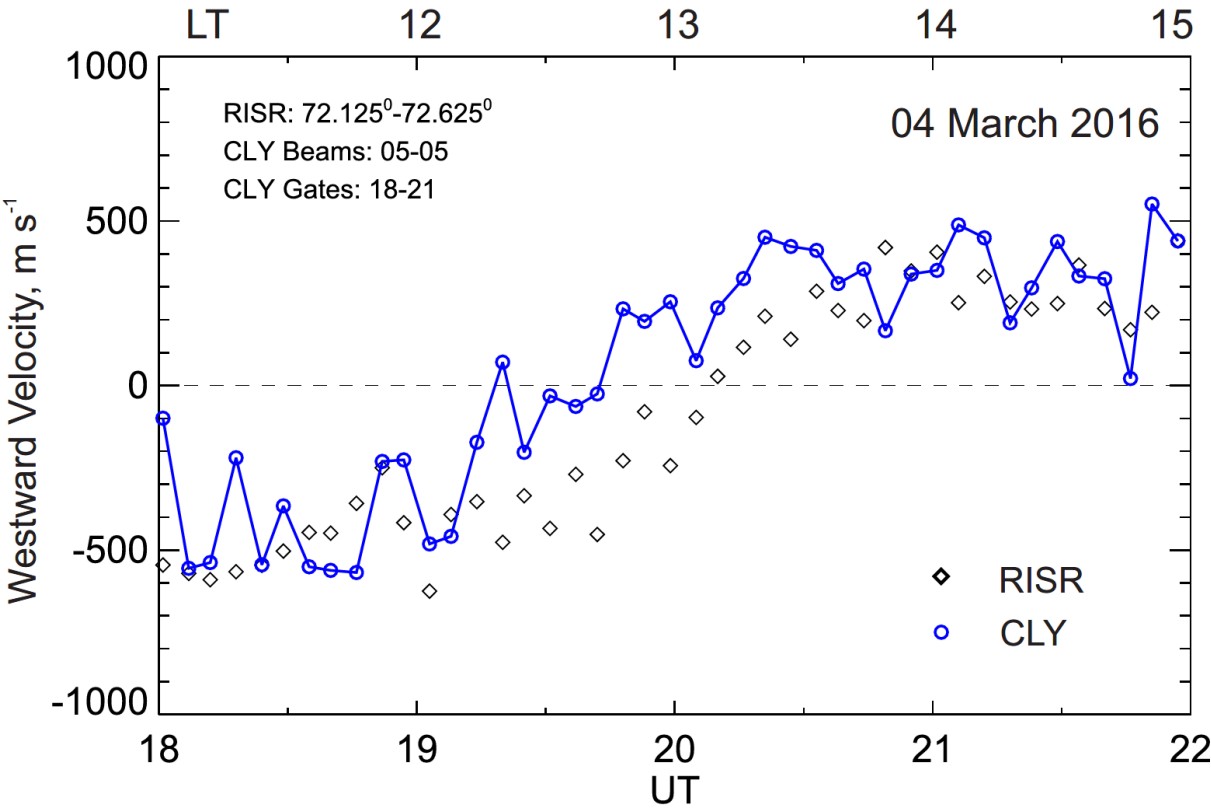

**Figure 5:** Eastward component of the **E×B** drift as measured by RISR (diamonds, 5 min resolution data) and matched velocity medians of CLY observations (blue circles, 5 min medians of original 1 min measurements in beams/gates "overlapping" the region of RISR observations) for the event of 04 March 2016. For the area of observations, local time (scale at the top) roughly coincides with the magnetic local time.

Figure 6a illustrates the typical spatial velocity distribution within the radar FoV, for one velocity scan during the above event. A sharp change in the LOS velocity polarity in the poleward and equatorward portions of the FoV is noticeable. The polarity transition occurs in the central beams 5-7. Figure 6b gives a global-scale map of plasma flow inferred from all SuperDARN radar measurements. The flow pattern in Fig. 6b was calculated by applying the new SuperDARN statistical model by Thomas and Shepherd (2018) which became available just recently. The map has a number of vectors originating from the RKN and INV radar measurements as well as those from CLY measurements. The presence of highly curved flows is evident near noon. Under these conditions, both SuperDARN and RISR can have difficulties in the construction of a 2-D vector field.

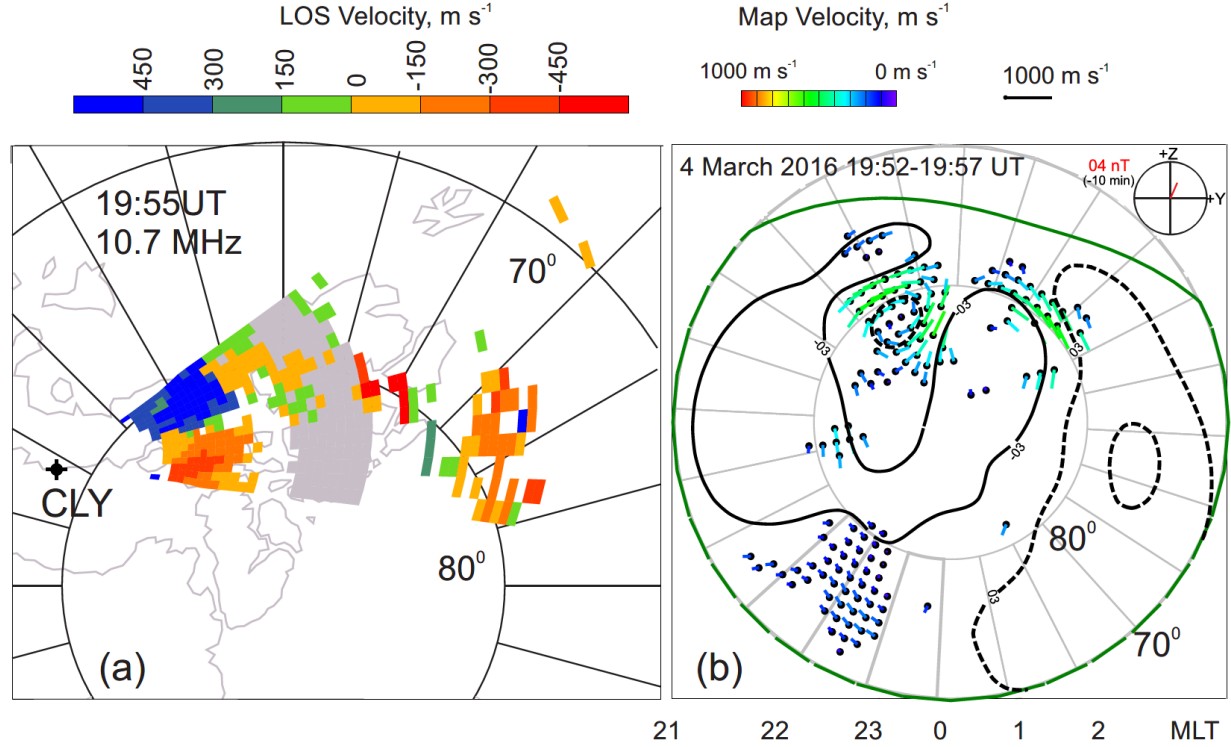

**Figure 6:** (a) A CLY LOS velocity map at 19:55 UT on 04 March 2016 and (b) a 5 min convection map calculated from all SuperDARN radar measurements for the same period of time. The TS2018 statistical model (Thomas and Shepherd, 2018) of the order 8 for the solar wind electric field of 2.1 mV/m was used. Contours of the electric potential are 6 kV apart.

We comment that the flows seen in Fig. 6 are sunward, roughly along the magnetic meridian near noon, signifying the occurrence of a reverse convection cell. This is expected since the IMF $B_z$ was steady at about +4 nT starting from 18:30 UT all the way until ~22 UT for this event.

Evaluating the extent that the SuperDARN and RISR vectors are affected by the shear in the flow is difficult. We can see that the centers (foci) of the convection cells, according to RISR and SuperDARN, do not coincide in latitude for many maps in this event.  In addition, the agreement between the RISR and SuperDARN map data improves dramatically when only  the lower latitude SuperDARN map data are considered.

We investigated this further by determining the location of the convection reversal boundary (CRB) for the reverse convection cell (like that shown in Fig. 6b by the dashed  contour). This is done by considering the standard 2 min SuperDARN maps, CLY LOS velocity maps, and

by looking at the reversal in the latitudinal profile of the RISR velocity (these are given for 5 min intervals). The CRB location based on the SuperDARN maps was determined by finding the middle latitude between the two neighbouring points on a standard plasma flow map with opposite directions of the flow, toward the Sun and away from the Sun. The CRB location based on the CLY LOS velocity maps was determined by plotting the LOS velocity versus beam number and

finding the azimuth and the range of the point at which the LOS velocity is zero. The CRB location from the RISR measurements was found by plotting the azimuthal component of the RISR plasma flow versus latitude and finding the latitude with zero velocity. All the CRB locations were given in terms of the geomagnetic latitude. The accuracy of the CRB determination in all cases is on the order of half of a degree of geomagnetic latitude.

The resulting data are presented in Fig. 7. The CRB inferred from SuperDARN maps is located almost $2°$ higher in MLAT than that determined from both CLY velocity maps and RISR data at the beginning of the event, and the differences are minimal toward the end of the event. The fact that the CRB location from CLY velocity maps is closer to that inferred from RISR data hints that perhaps the SuperDARN fitting procedure is the major factor for strong differences

between the SuperDARN maps and RISR measurements in this specific event. This is not to say that RISR measurements are exact; they are very likely also subject to errors under these strongly sheared and curved flows. One reason could be that the solution for the 2-D velocity vector field from the original LOS RISR data (Heinselman and Nicolls, 2008) smooths out the true sharp changes of the flow. Having a wider FoV for the RISR radar is expected to improve the quaity of

the flow pattern derivation under such conditions.

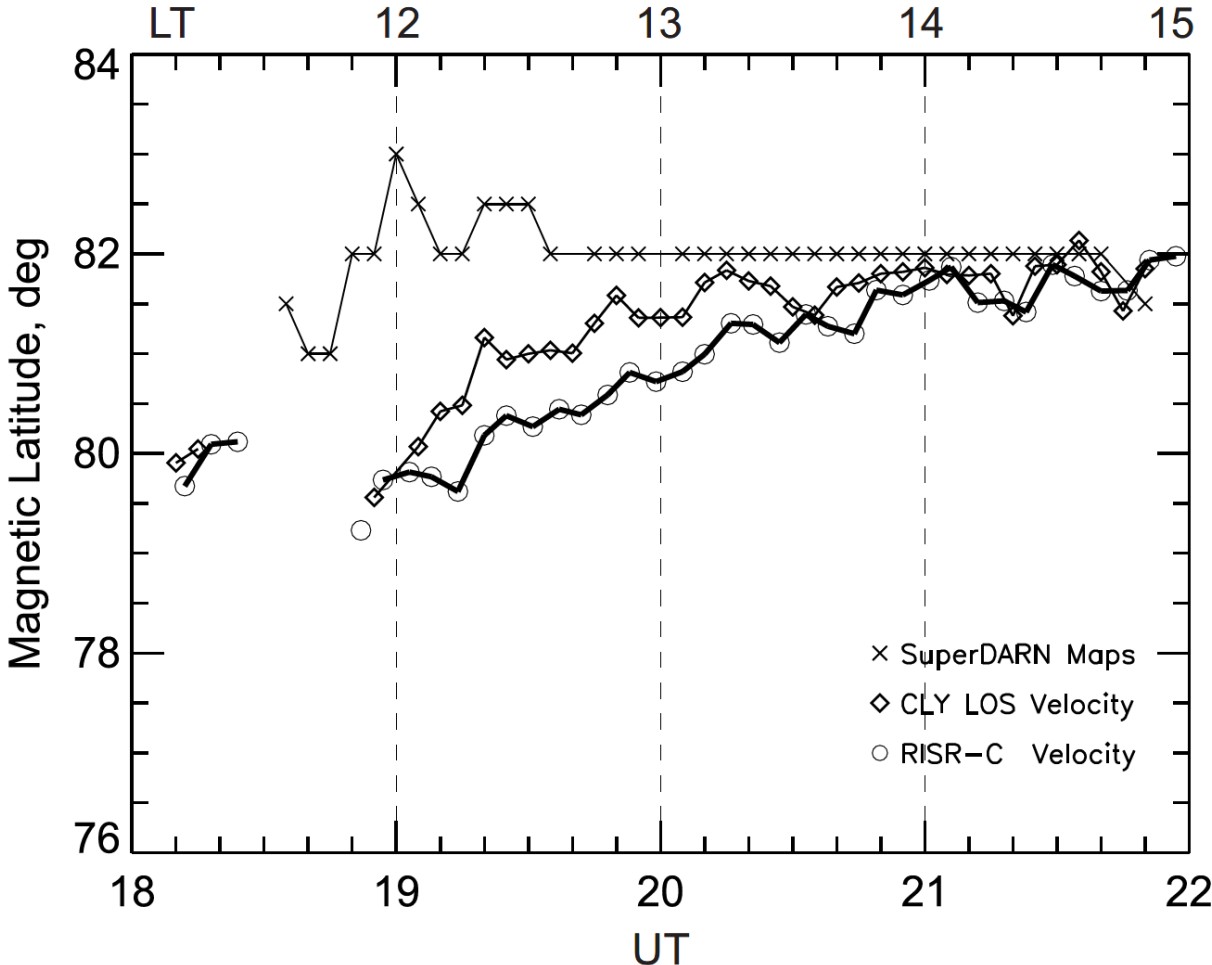

**Figure 7:** Magnetic latitude of the flow reversal location within the dayside reverse convection cell as inferred from SuperDARN convection maps (crosses), CLY LOS velocity maps (diamonds) and RISR measurements for the event of 04 March 2016. For the area of observations, local time (scale at the top) roughly coincides with the magnetic local time.

## Summary and conclusions

In this study, we attempted to validate the CLY SuperDARN radar velocity measurements by comparing them with the data collected by the Resolute Bay incoherent scatter radar (RISR). Because no line-of-sight velocity comparison is possible for the geometry of joint observations,

we adopted here a different approach. Namely, we considered the eastward component of $\mathbf{E} \times \mathbf{B}$ flow vector, as inferred from RISR measurements in multiple beams and compared it to CLY velocities from a number of eastward oriented beams and with the eastward component of the plasma flow inferred from 2-D SuperDARN maps. The analysis undertaken allows us to draw several conclusions.


1. The CLY radar velocities measured in beams 4-6 are statistically comparable to the $\mathbf{E} \times \mathbf{B}$ component of the plasma drift along these beams (eastward/azimuthal plasma flows) as measured by the RISR incoherent scatter radar. This implies that the velocity data provided by the CLY radar to the SuperDARN database are reliable and suitable for convection mapping involving all SuperDARN radars. The comparisons performed are an addition to the previous validation work for the RKN and INV SuperDARN radars.


2. The slope of the best linear fit line to the CLY velocity variation versus $\mathbf{E} \times \mathbf{B}$ component (as measured by RISR) applied to the binned values is on the order of 0.65 if all the available data (removing data with obvious ground-scatter contamination) in the range $\pm 1000$ m s$^{-1}$ are considered. Correction of HF velocities on the index of refraction effect improves the slope to ~0.75. The slope of the linear fit line for the corrected data is still below 1, implying that additional factors affect the relationship. Additionally, diurnal variations of the ratio of HF velocity to the RISR velocity shows their strongest decrease below one during nighttime but not daytime. This implies that the deterioration of RISR-SuperDARN velocity agreement at nighttime is caused not by the index by refraction effect but by other factors.



3. The effect of HF velocity underestimation for the CLY radar becomes progressively stronger for plasma drifts faster than about ~750 m s$^{-1}$.


4. One factor that may contribute to slower HF velocities, in addition to the refractive index, is the nature of HF signal collection. HF radars receive stronger signals from ionospheric regions with enhanced electron density where the electric field/$\mathbf{E} \times \mathbf{B}$ plasma drift can be decreased compared to the background plasma.


5. In a case of highly sheared plasma flows, such as near dayside reverse convection cells occurring under strongly dominant IMF $B_z > 0$, the differences between RISR and SuperDARN velocity vectors can be large.


6. The reasonable agreement between the velocities of the two systems quantified as the slopes of the linear fit lines at the level of 0.6-0.8 for both the LOS and 2-D comparisons, implies that the RISR technique of the $\mathbf{E} \times \mathbf{B}$ derivation from multiple individual radar beams is usually a reliable method most of the time. The comparison suggests that the RISR vectors are less reliable in the

midnight sector where the flows are often very irregular, and strong vertical motions occur.

**Acknowledgments**

Continuous funding of SuperDARN radars by National Scientific Agencies of Australia, Canada, China, France, Italy, Japan, Norway, South Africa, United Kingdom, and the United States of America is appreciated. The current research would be impossible without ongoing support from the Canadian Foundation for Innovation, Canadian Space Agency's Geospace Observatory (GO Canada) continuation initiative to the U of Saskatchewan radar group and NSERC Discovery grant

to A.V.K. The University of Calgary RISR-C radar is funded by the Canada Foundation for Innovation and is a partnership with the US National Science Foundation and SRI International. We thank C. Graf for the initial analysis of RISR density data and R. Fiori for the help in software development. Discussions of various aspects of the paper with P. V. Ponomarenko and his help in software development are appreciated. The authors are indebted to two anonymous reviewers who

not only identified weaknesses in the original manuscript but also made numerous constructive suggestions and, in addition, tremendously improved the writing style.

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

Author contributions.


RG and PB worked on raw data processing and their preliminary analysis. PB prepared some diagrams. AVK did most of the comparison work and wrote the initial manuscript. All authors participated in the writing, and all commented on the paper.