# Peer review of "Validation of Clyde River SuperDARN radar velocity measurements with the RISR-C incoherent scatter radar"

_Annales Geophysicae, 2018_

## Referee Comment (RC1) · Anonymous Referee #1 · 29 Oct 2018

The article 'Validation of Clyde River SuperDARN radar velocity measurements with the RISR-C incoherent scatter radar' presents a study that validates SuperDARN velocities measured from the Clyde River radar using the RISR-C incoherent scatter radar as a 'truth' data set. To my knowledge, this radar has not previously been validated, making this study a valuable contribution to the literature. The validation methods used are mostly appropriate, but could benefit from further clarification. The presentation and language are mostly clear, but have room for improvement (especially with respect to the use of articles). The length of the paper is adequate. The authors give proper credit to related work and relate how the study they present builds upon past validations. I believe this study will make an important scientific contribution once revisions have

been made.

**Title and abstract:** The title is clear and appropriate. The abstract needs some clarification on line 14 "...SuperDARN convection maps (constructed for the area of joint measurements) shows the effect of smaller HF velocities even at smaller velocities". The first part of this phrase begs the question: "Which other radars were used?". While it may not be necessary to name the other radars in the abstract, this phrase could be reworded to emphasis how this combined data is useful in a validation of a single radar. The second part of this phrase is difficult to read (is it saying that HF velocities are notably smaller than ISR velocities at times when the ion drifts are both large and small?), and should be reworded.

**Major Questions:**

1. Which background model was used to produce the mapped SuperDARN velocities?

2. The discussion regarding the linear fit would be improved by providing the R-squared value instead of just the slope of the fitted line (discussion on Line 167, 208 and elsewhere).

3. Why not used the cleaned data set, with groundscatter contamination removed, to produce the SuperDARN maps? This would improve the significance of the mapped study results.

4. Why is a 2 min convection map shown in Figure 6b? It would be more appropriate and informative to use the 5 min map used in this study.

5. In the paragraph starting on line 335, the detection of the convection reversal boundary (CRB) is mentioned. How is the CRB identified in the SuperDARN data and the RISR data? How accurate is each detection method?
6. How much data goes into each point in Figure 7?

**Minor Questions and Clarifications:**

1. What is meant by "above the E×B component" on line 39?

2. Since the beam centres are shown in Figure 1, shouldn't the shaded region be wider (encompassing the full width of all three beams)?

3. Paragraph on lines 131-135 does not make it clear how many median values of SuperDARN velocities are calculated. Is just one velocity median produced for each 5 minute × 3 beam × 4 gate bin? Or are up to 12 medians calculated in each 5 minute period?

4. Due to use of local time in discussion Figures 2, 4, 5, and 7 should plot local time on the x-axis instead of (or as well as) UT. This could be added on the upper x-axis.

5. Figure 2 captions says the figure plots the number of joint observations for all events. Does this include or exclude times with mixed ground scatter? Wording in discussion made this unclear.

6. The discussion of Figure 3 states that the black-white dots are close to the line of perfect agreement, and this seems to be true when there are more points. It would be useful to know, quantitatively, when the agreement degrades with respect to both the velocity magnitude and the number of velocity pairs.

7. "...data are spread across the local times, and so the R medians are dominated by low velocity data in (which bins?) of Figure 4." on Line 253.

8. On line 290, what do the data agree with? Each other?

9. Line 345: What types of errors are likely to be encountered? Is the limited RISR FoV important?

10. Point 2 of the summary: expand on the last sentence, spelling out the import of the results presented in the last sentence.

11. On line 390, what is meant by "strong IMF $B_Z > 0$"? Does it mean any IMF with $B_Z > 0$? Or a positive $B_Z$ with a large magnitude? If so, what magnitude is considered large?

12. Quantify what is meant by "reasonable agreement" on line 393.

13. Point 6 of the summary should be discussed better in the article's main text.

**Figure legibility:**

1. The pink asterisks and black bars in Figure 3 are not legible. I recommend outlining the black bars in white (as was done with the points). The same could be done for the pink asterisks, or they could be made larger or removed if the plot is too busy.

2. Label needed for colour bar on Figure 3.

3. Suggest changing velocity scale to $\pm$ 1000 $ms^{-1}$ in Figure 5

4. Colour bar needed for electric potential in Figure 6b, or remove colour contour.

5. In Figure 6, specify LOS and mapped velocity in colour bars.

**Grammar and organisation:**

1. HF needs to be defined on line 20

2. "...measure the Doppler velocity..." on line 22

3. Reword sentence ending on line 24 with "...plasma E×B drift."

4. "...(LOS) velocities and the E×B drift..." on line 30.

5. Remove commas flanking "published so far" on line 32.

6. Swap "accepted" and "now" on line 33.

7. "One factor found to lead to this result is an assumption made during SuperDARN velocity calculations, which sets the index of refraction for the ionosphere to unity." On lines 34-35, or reword in another fashion.

8. "...velocity magnitudes are substantially smaller..." Line 37.

9. replace "of >" with "exceeding" on line 38.

10. remove "often" on line 38.

11. "...are often above..." on line 39.

12. recommend replacing "Some" with "Other" on line 41.

13. "Despite obvious progress in measurement interpretations, HF-based..." on line 44

14. "...investigation to continue improving the quality..." on line 45.

15. "...unit is necessary to be confident in the reliability..." on line 47.

16. recommend replacing "work" with "study" on line 49.

17. "...this effort complements the previous validation..." on line 50.

18. "Since the CLY radar currently provides a significant contribution to the..." on line 53.

19. "We take advantage of the availability of E×B drift measurements made by the recently installed ISR RISR-C..." on lines 54-55.

20. RISR needs to be defined on line 55.

21. comma needed after "e.g." everywhere in text (e.g., line 56)

22. recommend placing the "e.g., Gillies et al. (2016)" reference in parenthesis on line 56.

23. Need to reword last sentence of paragraph ending on line 57, and move this sentence to the beginning of the following paragraph. Suggest something like: "In the present work we compare CLY and ISR-based velocities in a different way than previous studies."

24. "...systems that make measurements..." on line 58.

25. "...same direction are performed (e.g., Gillies et al., 2018)."

26. "...geometry due to the distance between the radars' beams" on line 60.

27. Recommend referring to Figure 1 in paragraph encompassing Line 60 instead of Figure 1 from Gillies et al. (2018). If more information is needed that is not included in this paper's Figure 1, I recommend adding it.

28. "...compare them with CLY data averaged over 3 beams and 4 gates." on line 64.

29. "...the average and median velocities..." on line 65.

30. "A validation using highly-averaged data is appropriate since the SuperDARN global-scale..." on line 66.

31. "...LOS velocities (the so-called gridded velocities)." on line 68.

32. "...up to 27 LOS velocity values in bins consisting of data within $\pm$ one radar..." on line 69.

33. "...implies that the input to the Potential..." on line 71.

34. "...space domain. Thus, a validation of the CLY contribution to SuperDARN convection maps can be performed using 2D RISR-C data and HF velocity..." on lines 72-73.

35. "...value in this CLY-RISR comparison." on line 76.

36. "...estimations also has some limitations (HN, 2008) that need testing. A couple of the limitations we will consider are a lack..." on lines 76-78.

37. "...SuperDARN is expected, but the degree of this agreement is not yet known." on lines 81-82.

38. "The approximate points where..." on line 92.

39. "...assigned are show in Figure 1 for the height of 300 km." on line 93.

40. Figure 1 caption: recommend using ISR instead of "incoherent scatter radar"

41. "...CLY beams 4, 5, and 6 (their centers) and the area where data were considered. The shaded region flanked by beams 4..." on Lines 105-106.

42. "...that within these range gates the CLY beams 4-6 are..." on line 108.

43. "...geographic latitude, as shown in Fig. 1." on Line 109.

44. "...CLY LOS velocities with the..." on Line 110.

45. recommend cutting "as given by RISR" on line 111.

46. "RKN and INV radars, so that..." on line 112.

47. "...comprising of about 1,000 h of RISER observations made over the entire year of 2016. The radar..." on lines 116-117.

48. Reword sentence on line 119.

49. "...available for winter and both equinoxes, with no measurements made in the summer. We consider 5 min RISR data, because they have much smaller measurement errors than the 1 min resolution data." on lines 121-123.

50. "...radar measurements, times when RISR and CLY both made measurements in the blue and shaded region shown in Figure 1, for various UT." on line 125.

51. "...because of the preferential..." on line 126.

52. "...interest during the daytime..." on line 127.

53. "...comparison. Periods when CLY data were contaminated by ground scatter were dropped from further consideration due to their profound affect on the velocity comparison (Gillies et al., 2018)." on Lines 128-130.

54. Recommend moving paragraph starting on 131 to be the second paragraph in Section 3.

55. "...select a 5 min period..." on line 131.

56. recommend placing "see blue circles in Figure 1" in parenthesis on line 132.

57. "...compute a median.." on line 133.

58. "...velocity over a matching 5 minute interval in 3 beams..." on line 134.

59. "Data binned in this way are shown by..." on line 149.

60. "...are almost linearly related, especially..." on line 164.

61. "...the bisector are the RISR velocities with magnitudes greater than..." on line 165.

62. Reword last sentence on paragraph containing line 165.

63. "...LOS comparison, shown in Figure 1." on line 173.

64. "We selected the three grid nodes at 81.5° magnetic latitude that were closest to the area of the CLY LOS velocity assessment and the two closest grid notes at 80.5° magnetic latitude, marked by red crosses in Figure 1." on lines 174-176.

65. "...location, the geographic Eastward component...." on line 176.

66. "...1 measurement) was calculated to represent the eastward plasma flow component of a 5 min SuperDARN map. This is not a traditional temporal..." on lines 178-181.

67. "...intervals often occurred at irregular times, while the SuperDARN maps were produced at exact..." at Line 184.

68. "10-15 min, and so on. For the comparison, only..." on line 185.

69. "...component was usually available at all locations shown..." on line 189.

70. space missing on line 191 ("of0.25")

71. Replace line 193 with "The selection criteria produced a slightly smaller (but still statistically significant) data set than was obtained for the LOS..." on Lines 193-194, removing the "Obtained data pairs..." sentence.

72. Consider adding something like ", with the SuperDARN vectors calculated using measurements from CLY, RKN, and INV, as well as the XXX model." to the end of the last paragraph of section 5.

73. "...plasma flow measured by RISR and SuperDARN. The data..." on line 200.

74. "...Figure 3b using the same methods as performed on Figure 3a, see Section 4." on lines 201-202.

75. "...SuperDARN is expected." on line 214.

76. "...of 300-600 $ms^{-1}$. The inconsistencies are characterized by slower Super-DARN velocities." on Line 216.

77. "...measurements, giving..." on line 220.

78. "One reason frequently given for the systematic 'underestimation' of the Super-DARN velocity measurements is the assumption that the index of refraction is unity (e.g., Gillies et al, 2009; Ponomarenko et al. 2009)." on lines 225-27.

79. "..., previous studies, though it does not entirely account for the differences between the radar measurements. (start new paragraph) . We also investigated the..." on line 234.

80. "...velocity ratio R=$v_{HF}/v_{RISR}$ as done previously by Gillies et al. (2018) to explore possible influences of the refractive index on velocity using typical local time variations in electron density as a proxy for refractive index. For the winter and equinoctial ionosphere..." on lines 235-238.

81. "...are typically observed near local solar noon and the afternoon..." at line 239.

82. "...smallest at these times, as reported by..." on line 241.

83. "...not as far below unity as near..." on line 244.

84. "...likely, incorrectly estimated for..." on line 246.

85. "...RB, we expect that this effect will also..." on line 247.

86. "...significantly. It is lower in the daytime...than during dawn/pre-noon hours (12-18 UT), but its values are..." on Lines 249-250.

87. "...This is probably because the infrequent..." on line 252.

88. "...R values are caused by an 'overestimation' of the true..." on line 254.

89. "...LOS velocities. Our data..." on line 257.

90. "...enhancements and decreases affect..." on line 277.

91. "...by ISRs. Such points are occasionally seen in..." on lines 288-289.

92. "... data (e.g., Ruohoniemi et al. 1987; Davies et al., 1999). Our data in Figure 3 also..." on lines 289-290.

93. cut "just for one CLY velocity scan" on Line 314.

94. "...within the CLY FoV, for one velocity scan during the above event." on Line 315.

95. "...originating from the RKN..." on line 319.

96. "The presence of highly-curved flows is evident near noon." on line 320.

97. Figure 6 caption: "...a standard 2 min convection map calculated from..."

98. Remove "highly" on line 327.

99. "Evaluating the extent that SuperDARN and..." on line 330.

100. "In addition, the agreement between the RISR and SuperDARN map data improves dramatically when only the lower latitude SuperDARN map data is considered." on lines 332-334.

101. "We investigated further by determining the location..." on line 335.

102. "...like that shown in the Figure 6b contours). This is done by considering..." on line 336.

103. "...CLY velocity scan maps, and by..."

104. Use same language in Figure 7 caption as in the discussion.

105. "...we adopted a different approach. We considered..." lines 359-360.

106. "...flow vector, as..." line 360.

107. "...multiple beams, and compared it to the CLY velocities from a number of eastward oriented beams and wit the..." on lines 360-362.

108. "...radar velocities measured in beams 4-6 are statistically..." on line 366.

109. "...below one, implying... on line 377.

110. "...diurnal variations of the ratio..." on line 378.

111. "...velocity show their strongest..." on line 379.

112. "...stronger for plasma drifts faster than about 750 $ms^{-1}$." on lines 381-382.

113. "One factor that may contribute to slower HF velocities in addition to the refractive index is the nature of HF propagation. Because HF radars receive stronger signals from..." on lines 384-385.

114. "...radar beams is usually a reliable method. The comparison..." on lines 394-395.

---

## Referee Comment (RC2) · Anonymous Referee #2 · 1 Nov 2018

Review of "Validation of Clyde River SuperDARN radar velocity measurements with the RISR-C incoherent scatter radar" by Alexander Koustov, Robert Gillies and Peter Bankole

This paper is the latest of a long series of papers that compare HF and IS velocity measurements with the aim of ascertaining whether F-region HF velocity measurements are representative of the ExB drift – and if not, why not. This endeavor is both interesting and useful, and this paper makes a solid contribution to that body of work. The paper is generally clear - although slightly awkward in places - and concise. The figures are clear and attractive, and supportive of the conclusions reached. In the ref-

eree's view, the paper is well worthy of publication, preferably once a few minor comments have been taken into consideration (just to ease readability of the manuscript).

Line 13 : "eastward plasma flow component" instead of "plasma flows eastward component" Line 14 : "effects of smaller HF velocities even at smaller velocities" is a little unwieldy Line 15 : suggest "...differences in eastward velocities between the two instruments..." Line 23 : is it worth mentioning here that echoes are not only from the F-region? Line 27 : Davies et al., 1999 (not Davis) Line 27 : Perhaps worth adding Davies et al. (2000) Annales Geophysicae, 2000, 18 (5), pp.589-594 Line 28 : Sentence starting "These observations...". Please clarify what is meant by the "Super-DARN principle in plasma flow measurements." You mean that F-region irregularities travel roughly with the E x B flow? Line 32 : Collecting area was only one of a few suggestions for the discrepancy proposed by Davies et al. (1999) Line 33 : "It is accepted now that the HF velocities *in the F-region* are generally smaller (Gillies et al., 2018)." Line 34 : "SuperDarn measurements are interpreted under the assumption" is perhaps clearer Line 37 : "HF velocity magnitudes are" Line 44 : "Despite obvious progress" Line 49 : "Undertake validation work" Line 54 : "We take here advantage of the availability of E×B" Line 56 : "An important aspect of the present work is that we compare CLY and ISR-based velocities in a different way as compared to the previous studies..." Line 60 : "none of these radar's beams are close enough *in terms of their direction*" Line 65 : "spatial domain" Line 69 : Sentence starting "These are inferred..." would benefit from clarification. Line 75: "...value from the..." Line 76 : "...The RISR method of velocity vector estimations..." Line 80 : "...both radars'..." Line 81 : remove highly Line 88 : hereafter (rather than starting from here) Line 92 : perhaps use "bin size" instead of step throughout Line 93 : "The points to which the measurements are assigned." Fig caption 1 : The black straight lines are the orientation of specific beams (4-6 for CLY), data from which are investigated. Fig caption 1 : The solid red arcs are the magnetic latitudes of 75°, 80° and 85°. Line 105 : "(along their centers) and the area *from which* data were considered," Line 110 : eastward Line 118 : "On the days when the radar was operational it typically worked for the

whole 24 hours, albeit switching, once-in-a-while, its mode..." Line 124 : "...also available." Line 125 : "...measurements as a function of UT. The number of intervals..." Line 125 : perhaps "from noon to dusk" rather than during noon-dusk time. Line 130 : perhaps it is worth clarifying the means by which ground scatter has been identified for removal. Line 131 : "...select a 5-min..." Line 132 : geographic latitudes? Line 133 : "and compute the median..." Line 134 : "..., as mentioned above." Fig caption 2 : "for all data considered" perhaps? It may be clearer to use "data points" than "data intervals", even though they are 5-mins long... Line 146 : "Although some spread is present, a significant number of points are located..." Line 146 : Instead of "bisector of perfect agreement", perhaps use "line of equality" throughout(or 1:1 line). Line 148 : "...we binned the CLY data according to the RISR measurements, using 100-m/s bins of the latter." Line 149 : "Data binned in this way are shown..." Line 149 : Median? Fig caption 3 : The standard deviations are not very clear on the plot. Why use median and standard deviation, not median and quartiles? Fig caption 3 : "...but the eastward flow component" Line 163 : "Good alignment with the bisector and good correspondence between the location..." Line 167 : "If one *describes* the dependence by a linear fit line between velocities of +/- 1000 m/s, the slope of the *fitted* line is ∼0.65." Line 172 : "We restrict consideration..." Line 172 : "Here the SuperDARN convection vectors are available at geomagnetic latitudes of 80.5° - 81.5° and ∼ 7° of magnetic longitude." This is the same region as for the LOS comparison, right? It is worth mentioning here what SuperDARN radars are included, just to ensure that it is clear it is not just CLY data. Line 175 : "grid node locations" Line 177 : eastward (not astward) Line 180 : "because the RISR" Line 182 : "(which is usually 2 min)" Line 184 : "the start times of RISR measurement intervals were often irregularly spaced, while the SuperDARN maps were synchronized to exactly correspond to 5-min boundaries (i.e. 0-5 min, 10-15 min, etc). " Line 186 : "For comparison, only HF and ISR data that were less than 2 min apart were considered." Line 189 : "For RISR, the eastward E× B velocity component is usually available at all points shown by open circles in Fig. 1." Line 193 : "All obtained data pairs were entered into a common dataset." Line 201 : "Figure 3b

plots eastward component of the plasma flow from the two systems. The spread of the data. . ." Line 202 : "We assessed Fig. 3b by binning the data in the same way as for to Fig. 3a, see open circles." Line 202 : "Several results from Fig. 3b are consistent with the data of Fig. 3a. Line 207 : "If one describes the dependence by a linear fit to the velocity medians in bins (open circles) between +/- 1000 m/s, the slope of the fitted line is 0.54." Line 208 : "Secondly, the tendency for the SuperDARN velocity being smaller is greater for larger RISR magnitudes." Line 208 : "show opposite velocity polarity" – "show oppositely directed flows", perhaps? Line 212 : "Although Fig. 3 shows good consistency of the data provided by the two radar systems, the differences can be as large as a factor of 2 in individual measurements." Line 214 : delete "highly" Line 219 : "broader area over which the SuperDARN data are averaged for the 2-D comparison" Line 220 : "In this case, there is more chance for SuperDARN to include ground-scatter. . ." Line 227 : One popular opinion about SuperDARN velocity measurements is that the systematic "underestimation" of the velocity by the HF radars is due to the index of refraction being assumed to be unity." Line 227 : "A plot similar to Fig. 3a" Line 232 : "The plot looked very similar to Fig. 3a." Line 234 : what does "except the slope is not quite close to 1" mean? Other studies has a slope closer to 1? Line 236 : ". . .plotted R from the RKN SuperDARN radar against UT" Line 239 : ". . .observed near local solar noon and during the afternoon hours. . ." Line 240 : "It is therefore expected that the velocity ratio R would be smallest during these times. . ." Line 240 : " (but not as strong as they were near noon)" Line 246 : "incorrectly" not "not correctly" perhaps Line 250 : "It is lower during daytime (noon is at about 19:00 UT) than during dawn/prenoon (12-18 UT). . ." Line 264: Please clarify the comment about lateral deviations of CLY beams giving both smaller and larger deviations. Line 275 : "in THE case of A" Line 276 : ". . .smaller than in the case of a uniform. . ." Line 278 : "The RISR radar would average the velocity in patches of enhanced. . . " Line 278 : Do you mean average the velocity in those patches together (not equally)? Line 283 : "decreased, so that they would show" Line 285 : "the opposite" Line 288 : "Such points are occasionally seen. . ." Line 290 : "Figure 3 also shows such points but, in general,

the data agree fairly well" Line 290 : "Although the work of Kustov..." Line 291 : clarify what is meant by "the effect" Line 292 : "...it could partially be due to the aforementioned effect of ionospheric..." Line 296 : "mixed ionospheric and ground scatter" Line 300 : "have to remind the reader that..." Line 300 : obvious period of ground scatter have been removed : again, what thresholds are used. Line 300 : presumably obvious ground scatter is removed from all figs... Fig 5: Are the diamonds really red? Line 314 : "one CLY scan" Line 315 : "during the above event." Line 315 : perhaps in the poleward and equatorward portion of the FOV, rather than high and low number beams. Line 319 : originating instead of originated Line 319 : "as well as those from CLY..." ? Line 320 : "near noon", rather than "at near noon hours" Fig 6 : convection map (not maps) Line 326 : "...are sunward, roughly along the magnetic meridian, near noon, signifying..." Line 327 : remove highly Line 330 : "the extent that" Line 331 : "We can see that the centres..." Line 331 : many cases? Many maps, do you mean? Line 335 : "To investigate this further..." Line 337 : "velocity scan maps" : maybe LOS velocity map (as used before) Line 339 : "the CRB inferred from SuperDARN maps is located almost 2 degrees higher in MLAT than that determined from both CLY velocity maps and RISR data at the..." Line 340 : "The fact that the CRB from CLY velocity maps is closer to that inferred from RISR data hints that the SuperDARN..." Fig 7: in various beams? Which ones? Line 353 : "Resolute Bay incoherent scatter radar (RISR)." Line 360 : "approach. Namely, we considered the eastward component of the ExB..." Line 364 : "draw several conclusions" Line 266 : Conclusion 1 should be combined with conclusion 6 as the "final conclusion". Line 279 : "nighttime, but not daytime." Line 384 : Could the RISR electron density measurements be used to test if this is true on a statistical basis? Line 386 : "compared to"

---

## Author Comment (AC1) · 23 Nov 2018

Reply to the referee #1

We thank this referee for carefully studying the manuscript, making numerous suggestions, improving the presentation style, and suggesting extensive corrections to our writing. We commend his/her truly amazing seriousness and dedication to the referee's duties. This is a job well done and way above any reasonable expectation! The lead author wishes all his papers were scrutinized the way it was done here.

Below is our reply on a point-by-point basis. All suggested grammatical changes by the Referee #1 are given in a red color. In the revised manuscript, all new statements made in response to both referees' comments are given by a green color.

**Title and abstract:** The title is clear and appropriate. The abstract needs some clarification on line 14 "...SuperDARN convection maps (constructed for the area of joint measurements) shows the effect of smaller HF velocities even at smaller velocities". The first part of this phrase begs the question: "Which other radars were used?". While it may not be necessary to name the other radars in the abstract, this phrase could be reworded to emphasis how this combined data is useful in a validation of a single radar. The second part of this phrase is difficult to read (is it saying that HF velocities are notably smaller than ISR velocities at times when the ion drifts are both large and small?), and should be reworded.

In SuperDARN research, it is traditional to use ALL available velocity data for the construction of every individual flow map. Although the clarification is not really needed we added it because, perhaps in the context of the study, the reader might think that the convection maps were built using only CLY radar data (this is conceivable but would be bizarre). We modified the second part of the statement.

**Major Questions:**

1. Which background model was used to produce the mapped SuperDARN velocities?

We used the traditional Ruohoniemi and Greenwald (1996) statistical model. This is now mentioned in the text, line 216. The new model by Thomas and Shepherd (2018) was NOT available to us at the time of the work being done. However, now it is in place. We reproduced Figure 6 with this new model, as we were asked to improve the original Figure 6. We added the reference on this new model as well (lines 572-574).

2. The discussion regarding the linear fit would be improved by providing the R-squared value instead of just the slope of the fitted line (discussion on Line 167, 208 and elsewhere).

We produced linear fits to the scatter points in Figures 3a,b and provided information on a number of points involved and $R^2$ values in newly introduced Table 1, see lines 175-188 and 239-245.

3. Why not used the cleaned data set, with groundscatter contamination removed, to produce the SuperDARN maps? This would improve the significance of the mapped study results.

The ground scatter contamination in SuperDARN mapping is an issue that has not been fully addressed. So far, NOBODY treats the mixed scatter, even for individual events and even for one specific radar because this is quite a tedious process. We used the standard criteria as described by Ponomarenko et al. (2007). We added a note on this (lines 136-137) and the reference (lines 546-548).

Why is a 2 min convection map shown in Figure 6b? It would be more appropriate and informative to use the 5 min map used in this study.

We replaced the 2 min map by a new one, obtained by using 5 min SuperDARN grid velocities.

4. In the paragraph starting on line 335, the detection of the convection reversal boundary (CRB) is mentioned. How is the CRB identified in the SuperDARN data and the RISR data? How accurate is each detection method?

We put more wording on this, lines 381-389. The accuracy is not great, on the order on 0.5º of MLAT.

5. How much data goes into each point in Figure 7?

We added description of how this Figure was built, lines 381-389.

**Minor Questions and Clarifications:**

1. What is meant by "above the E×B component" on line 39?

We are not sure what is confusing in here. It means that the HF velocity is larger than the ExB component of the plasma flow along the radar beam. We added words, line 43, green color.

2. Since the beam centres are shown in Figure 1, shouldn't the shaded region be wider (encompassing the full width of all three beams)?

We modified Figure 1 to indicate that the width of the beams is not zero.

3. Paragraph on lines 131-135 does not make it clear how many median values of SuperDARN velocities are calculated. Is just one velocity median produced for each 5 minute × 3 beam × 4 gate bin? Or are up to 12 medians calculated in each 5 minute period?

One median. We mention this now on line 129.

4. Due to use of local time in discussion Figures 2, 4, 5, and 7 should plot local time on the x-axis instead of (or as well as) UT. This could be added on the upper x-axis.

We added LT times on Figures 2, 4, 5, 7 as recommended.

5. Figure 2 captions says the figure plots the number of joint observations for all events. Does this include or exclude times with mixed ground scatter? Wording in discussion made this unclear.

The events with obvious ground scatter were not considered in Figure 2, it would be illogical to count them while the data were not actually used. We modified the text to make it clear, lines 134-137.

6. The discussion of Figure 3 states that the black-white dots are close to the line of perfect agreement, and this seems to be true when there are more points. It would be useful to know, quantitatively, when the agreement degrades with respect to both the velocity magnitude and the number of velocity pairs.

We made the linear fits and placed requested information in newly made Table 1. We also made some comments, see lines 175-188 and 239-245.

7. "...data are spread across the local times, and so the R medians are dominated by low velocity data in (which bins?) of Figure 4." on Line 253.

RISR bins. We modified the text, line 281, green color.

8. On line 290, what do the data agree with? Each other?

Yes, between each other. The text was modified as recommended by the referee #2, line 325.

9. Line 345: What types of errors are likely to be encountered? Is the limited RISR FoV important?

We added text stating that smoothing of the data and limited FOV are the potential factors, lines 397-400.

10. Point 2 of the summary: expand on the last sentence, spelling out the import of the results presented in the last sentence.

We added text, line 435-437.

11. On line 390, what is meant by "strong IMF Bz > 0"? Does it mean any IMF with Bz > 0? Or a positive Bz with a large magnitude? If so, what magnitude is considered large?

"Strongly dominant" Bz+ would be the correct word (Bz=+4 nT, By=+2 nT). We added this word, line 448.

12. Quantify what is meant by "reasonable agreement" on line 393.

Added, lines 451-452.

13. Point 6 of the summary should be discussed better in the article's main text.

We added more on fits and on potential problems with RISR (lines 397-400).

**Figure legibility:**

1. The pink asterisks and black bars in Figure 3 are not legible. I recommend outlining the black bars in white (as was done with the points). The same could be done for the pink asterisks, or they could be made larger or removed if the plot is too busy.

Modifications are made as recommended

2. Label needed for colour bar on Figure 3.

Added

3. Suggest changing velocity scale to ± 1000 $ms^{-1}$ in Figure 5

Modified as recommended

4. Colour bar needed for electric potential in Figure 6b, or remove colour contour.

Coloring of the electric potential contours was removed.

6. In Figure 6, specify LOS and mapped velocity in colour bars.

Added

**Grammar and organisation:**

Most of grammatical suggestions were accepted and modifications made, as shown by red color. Below we list only those that were not accommodated.

**First,** several suggestions that we ignored:

*21. comma needed after "e.g." everywhere in text (e.g., line 56*)

This is not a correct suggestion, Ann Geophys does not use comma, contrary to AGU journals.

*27. Recommend referring to Figure 1 in paragraph encompassing Line 60 instead of Figure 1 from Gillies et al. (2018). If more information is needed that is not included in this paper's Figure 1, I recommend adding it.*

We think that we handled the issue in a proper and best way. Adding RISR beam pierce locations or beam orientations to our Figure 1 (especially for the RISR imaging mode) would make the diagram very busy while Gillies et al. (2018) presented the plot exactly to address the issue.

*34. "...space domain. Thus, a validation of the CLY contribution to SuperDARN con vection maps can be performed using 2D RISR-C data and HF velocity..." on lines 72-73.*

We decided not to change because this replacement would completely distort our statement.

**Second,** we found that the referee #2 made a better suggestion for some statements, and we adopted his/her text. These are:

8. "...velocity magnitudes are substantially smaller..." Line 37.

26. "...geometry due to the distance between the radars' beams" on line 60.

35. "...value in this CLY-RISR comparison." on line 76.

38. "The approximate points where..." on line 92.

59. "Data binned in this way are shown by..." on line 149.

67. "...intervals often occurred at irregular times, while the SuperDARN maps were produced at exact..." at Line 184.

105. "...we adopted a different approach. We considered..." lines 359-360.

On behalf of co-authors, A. Koustov

[revised manuscript text omitted]

---

## Author Comment (AC2) · 23 Nov 2018

**Reply to the reviewer #2 comments**

First of all, we thank this reviewer for thoroughly going through the manuscript and making numerous valuable suggestions, they are all excellent. There is no doubt that this reviewer spent an enormous amount of precious research time reading and thinking about improvements of somebody's paper, and it is truly amazing. He/she perhaps did not realize that the number of suggestions exceeds 110. Perhaps this is a new standard for this journal. The lead author feels uneasiness as it would be difficult for him to write a review of somebody's paper at the same level of diligence and care.

Here is our reply on the comments. First, the vast majority of suggestions were accepted and corrections have been incorporated. Those are all shown by a blue color in the revised manuscript. Below we explain only those items that were corrected not exactly as was suggested. We note that reviewer #1 commented on several similar items, and we addressed some of them the way he/she recommended. Second, in response to some comments by both reviewers, new text was added; all these additions are colored in green.

Line 23 : is it worth mentioning here that echoes are not only from the F-region?
We feel that mentioning E region scatter will detour the reader from the main issue, velocity of F region echoes. We prefer not to go to discussion of E region echoes which is a big topic by itself.

Line 28 : Sentence starting "These observations. . .". Please clarify what is meant by the "Super-DARN principle in plasma flow measurements." You mean that F-region irregularities travel roughly with the E x B flow?
Yes, to clarify, we introduced a new statement with the word "assumption" so that the meaning would be obvious.

Line 32 : Collecting area was only one of a few suggestions for the discrepancy proposed by Davies et al. (1999)
Added statement on integration time, line 33-34.

Line 56 : "An important aspect of the present work is that we compare CLY and ISR-based velocities in a different way as compared to the previous studies. . ."
We prefer the wording given by referee #1

Line 69 : Sentence starting "These are inferred. . ." would benefit from clarification.
Modified, lines 72-73.

Line 234 : what does "except the slope is not quite close to 1" mean? Other studies has a slope closer to 1?
Corrected as suggested by Referee #1

Line 264: Please clarify the comment about lateral deviations of CLY beams giving both smaller and larger deviations.
Added in lines 290-296.

Line 300 : obvious period of ground scatter have been removed : again, what thresholds are used.
We added on lines 335-336 and on the criteria  - on lines  136-137.

On behalf of co-authors,                                    A. Koustov

[revised manuscript text omitted]